# Effects of Complex DNA and MVs with GTF Extracted from *Streptococcus mutans* on the Oral Biofilm

**DOI:** 10.3390/molecules24173131

**Published:** 2019-08-28

**Authors:** Hidenobu Senpuku, Tomoyo Nakamura, Yusuke Iwabuchi, Satoru Hirayama, Ryoma Nakao, Makoto Ohnishi

**Affiliations:** Department of Bacteriology I, National Institute of Infectious Diseases, Shinjuku-ku, Tokyo 162-8640, Japan

**Keywords:** biofilm, eDNA, *Streptococcus mutans*, GTF, membrane vesicles

## Abstract

*Streptococcus mutans* is one of the principal pathogens for the development of dental caries. Oral biofilms formed by *S. mutans* are constructed of insoluble glucan formation induced by the principal enzymes, GTF-I and GTF-SI, in sucrose-containing conditions. However, as another means of biofilm formation, extracellular DNA (eDNA) and membrane vesicles (MVs) are also contributors. To explore the roles of eDNA and MVs for biofilm formation, short and whole size pure DNAs, two types of sub-purified DNAs and MVs were extracted from *S. mutans* by beads destruction, treatment of proteinase K, and ultracentrifugation of culture supernatant, and applied into the biofilm formation assay using the *S. mutans* UA159 *gtfBC* mutant, which lost GTF-I and GTF-SI, on a human saliva-coated 96 well microtiter plate in sucrose-containing conditions. Sub-purified DNAs after cell lysis by beads destruction for total 90 and 180 s showed a complex form of short-size DNA with various proteins and MVs associated with GTF-I and GTF-SI, and induced significantly higher biofilm formation of the *S. mutans* UA159.*gtfBC* mutant than no sample (*p* < 0.05). Short-size pure DNA without proteins induced biofilm formation but whole-size pure DNA did not. Moreover, the complex form of MV associated with GTFs and short-size DNA showed significantly higher biofilm formation of initial colonizers on the human tooth surface such as *Streptococcus mitis* than no sample (*p* < 0.05). The short-size DNAs associated with MVs and GTFs are important contributors to the biofilm formation and may be one of additional targets for the prevention of oral biofilm-associated diseases.

## 1. Introduction

The formation of oral biofilms is the most important physical biological activity of oral bacteria and provides survival conditions for bacteria in the oral cavity when the physical removal of the bacteria is attempted by oral brushing and rinsing [1,2,3]. More than 700 species of oral microorganisms have been detected in the oral cavity, and some colonizers attach to the tooth surface, interact with each other, aggregate, and form complex bacteria biofilms [4,5,6]. In oral biofilm bacteria, *Streptococcus mutans* is a type of bacteria that attaches to the tooth surface and is one of the principal pathogens for the development of dental caries [7,8]. *S. mutans* has been shown to produce acids, exhibit a high degree of acid tolerance, produce bacteriocins, possess high-affinity systems for the assimilation of many carbohydrate sources such as glucan and fructan, and form sticky biofilms [9].

Water-insoluble glucan synthesized using GtfB (GTF-I)- and GtfC (GTF-SI)-glucosyltransferases, which are encoded by *gtfB* and *gtfC*, promote adhesion to tooth surfaces, aggregation of bacterial cells within the biofilm, and mature biofilm formation with complex bacteria in conditions supplemented with sucrose [10]. Surface-associated glucan-binding proteins (Gbps; GbpA, GbpB, GbpC, and GbpD) also mediate aggregation and biofilm formation by their affinity for glucan, thereby promoting plaque formation that contributes to the cariogenicity of *S. mutans* [11,12]. Furthermore, many other factors that may be responsible for biofilm formation have been reported: Aggregation by salivary agglutinins [13], extracellular DNA (eDNA) as the biofilm matrix [14,15], and dead cells [16]. Cell lysis provides bacterial adherence, aggregation, accumulation, and increasing biofilm biomass through the release of eDNA into the extracellular matrix [17,18]. Cell lysis is a vital process in releasing membrane vesicles (MVs) in gram-positive bacteria. MVs deliver virulence factors to host cells and induce immune responses and facilitate biofilm formation [19,20,21]. *Bacillus subtilis*, *Streptococcus pneumoniae*, *S. mutans*, and *Staphylococcus aureus* MVs contain several virulence factors, such as GTF-I and GTF-SI, GbpC, toxins, and peptidoglycan [19,22,23].

The insoluble glucan-independent biofilm of *S. mutans* was formed by increasing eDNA in primarily low pH conditions [24]. The degradation of eDNA by the addition of DNase I results in a significant decrease in biofilm formation [19,24]. eDNA is environmentally obtained from plaque in the oral cavity that is taken without targeting a particular species. Among eDNA, genomic DNA released from dead cells has important functions as an attachment factor for surfaces and an adhesive factor among biofilm bacteria during the initial stage of biofilm formation [15,25]. Recently, we reported that the presence of raffinose and small amounts of sucrose induced fructan synthesis by fructosyltransferase and aggregated eDNA into the biofilm in *S. mutans* [26]. Raffinose and a small amount of sucrose have cooperative effects, and this induction of biofilm formation depends on supportive elements, which mainly consist of eDNA and fructan.

To clarify the roles of eDNA, four types of DNA (short and whole sizes pure DNA, and two types of sub-purified DNAs) and MVs were extracted from *S. mutans* and applied into the biofilm formation assay using the *S. mutans* UA159 *gtfBC* mutant, which lost GTF-I and GTF-SI in sucrose-containing conditions. Sub-purified short size DNA associated with various proteins and MVs significantly induced GTF-dependent biofilm formation, but pure whole size DNA without proteins did not. In contrast, short-size pure DNA induced significant GTF-independent biofilm formation in *S. mutans* UA159 *gtfBC* mutant. This study suggests that short size DNAs associated with MVs and GTF are important for the formation of biofilms by pathogenic factors and may be one of the additional targets for the prevention of oral biofilm-associated diseases.

## 2. Materials and Methods

### 2.1. Bacterial Strains and Culture Conditions

Twenty eight strains in 17 bacterial species and two fungus species were used for experiments. *S. mutans* UA159, *S. mutans* UA159 *gtfBC* mutant (UA159.*gtfBC*^−^) [27], GS-5, MT8148, *Streptococcus sanguinis* ATCC10556, ST205, ST134, *Streptococcus gordonii* ATCC 10558, *Streptococcus mitis* ATCC 6249, ATCC 903, *Streptococcus oralis* ATCC 35037, *Streptococcus anginosus* ATCC 33397, *Streptococcus intermedius* ATCC 27335, *Streptococcus pyogenes* K32, K33, *Streptococcus pneumoniae* GTC 261, *Streptococcus salivarius* JCM5907, ATCC 9759, HT9R, *Actinomyces naeslundii* X600, *Actinomyces oris* MG1, *Neisseria flavescens* ATCC 13120, *Neisseria cinerea* 23-1, *Neisseria mucosa* 16-2, and *Neisseria subflava* #2 were maintained and grown in brain heart infusion (BHI) broth (Becton/Dickinson, Sparks, MD) at 37 °C in a 5% CO_2_ aerobic atmosphere (Gas pack: Mitsubishi Gas Chemical Co., Inc. Tokyo, Japan). *Candida albicans* SC5314, SC5312, *Staphylococcus aureus* Cowan I and ATCC 6538P were maintained and grown in BHI at 37 °C in an aerobic atmosphere. Tryptic soy broth without dextrose (TSB), Todd-Hewitt broth (THB), and BHI (Becton/Dickinson) with 0.25% glucose or 0.25% sucrose were used for biofilm formation, growth curves, aggregation assays, and observation of cell viability.

### 2.2. Extraction of Four Types of DNA

Four types of DNA (whole-size and short-size pure DNAs, and sub-purified DNAs after a total 90 and 180 s beads destruction) were extracted from *S. mutans* UA159 cultured in an aerobic atmosphere containing 5% CO_2_ (Gas Pack, Mitsubishi Gas Chemical Company) and BHI at 37 °C. For extraction of whole-size pure DNA, the cells were harvested by centrifugation at 6000× *g* after an overnight culture in 500 mL of BHI. The pellets were frozen overnight, and the next day, the pellets were suspended in lysozyme solution (2 mg/mL of lysozyme, 50 mM of glucose, 10 mM of cyclohexane diamine tetra acetate, and 25 mM of Tris-HCl, pH 8.0), incubated for 1 h and treated with proteinase K for 1 h at 37 °C. The full-size DNA was purified by a spin column in the DNeasy^®^ Blood and Tissue Kit (Qiagen, Venlo, Netherlands). As samples for sub-purified DNAs, the first (90 s destruction) and second types (180 s destruction) of subpurified DNA were extracted from whole cells harvested by centrifugation at 6000× *g* after an overnight culture in 500 mL of BHI. The pellets were frozen overnight, and the next day, they were suspended in lysozyme solution. The suspension was incubated at 37 °C for 1 h, and these samples were transferred to a 2-mL tube containing 0.5 g of 0.1-mm diameter glass beads. The suspension was shaken on a TissueLyser (Qiagen) for a total of 90 s (one set: 30 s of bead beating + 60 s cooling interval at 4 °C × three times) or 180 s (one set: 60 s of bead beating + 60 s cooling interval at 4 °C × three times) to homogenize the cells. These destroyed cells were centrifugally separated into solution samples and cell debris. The supernatants were collected for partial purification of the DNA. The subpurified DNA was precipitated with 70% (*v/v*) ethanol. The samples were washed with 70% (*v/v*) ethanol three times and used as DNA with proteins in subpurified DNA samples divided for a total of 90 s and a total of 180 s destruction. For the fourth type (short-size pure DNA) of DNA, we prepared DNA samples where we removed the proteins by phenol-chloroform extraction from the subpurified DNA (DNA associated with protein). Equal amounts of phenol:chloroform:isoamyl alcohol (25:24:1) were added to the subpurified DNA solution from the cells that were destroyed for 180 s. After centrifugation (4 °C, 14,000× g, 5 min), the aqueous phase was transferred into a 1.5 mL eppendorf tube. These procedures were repeated two times. After that, the DNA in the aqueous solution was precipitated with 70% (*v/v*) ethanol. The sample was washed with 70% (*v/v*) ethanol three times. To completely remove the proteins from subpurified DNA treated with phenol-chloroform, the subpurified DNA was treated with proteinase K for 1 h at 37 °C, precipitated, and washed with 70% (*v/v*) ethanol. Subpurified DNA treated with phenol-chloroform and proteinase K was extracted as short-size pure DNA and compared with whole-size pure DNA and two types of DNA with proteins in subpurified DNAs. Concentrations of DNA samples were measured by the NanoDrop Lite spectrophotometer. DNA in subpurified DNAs was visually analyzed to observe binding of MVs by a scanning electron microscope.

### 2.3. Extraction of MVs

*S. mutans* strains were cultured in 900 mL of BHI medium at 37 °C for 24 h. Preparation of MVs was performed as described previously with some modifications [28]. Culture supernatants were separated by centrifugation (6000× *g*, 20 min) and categorized as >10 kDa or >50 kDa. MVs (>10 kDa or >50 kDa) in supernatants were concentrated by centrifugal filters (Amicon Ultra 4, Merck kGaA, Darmstadt, Germany or VIVASPIN 20, Sartorius, Stone House, United Kingdom). Briefly, the concentrated MVs were filtered through polyvinylidene difluoride (PVDF) filter membranes (Merck kGaA) with pore sizes of 0.45 µm and 0.22 μm. The MV samples were ultracentrifuged (150,000× *g*, 2 h), and pellets were suspended as the first sample with sterile phosphate buffered saline (PBS; pH 7.2). In contrast, after ultracentrifugation, the supernatant was used as a culture supernatant sample without MVs. MVs were visually observed with a scanning electron microscope (Regulus 8220, Hitachi High-Technologies, Tokyo, Japan). The protein concentration of MV samples was determined using the Bio-Rad Protein Assay Kit (Bio-Rad Laboratories, Inc., Hercules, CA, USA). The double-stranded DNA (dsDNA) and RNA concentrations of MVs were measured by absorptiometry using a NanoDrop Lite spectrophotometer (Thermo Fisher Scientific, Waltham, MA, USA).

### 2.4. Human Saliva Collection

Whole saliva samples were collected from three healthy human volunteers (25–55 years old) after stimulation by chewing paraffin gum and pooled into ice-chilled sterile bottles over a period of 5 min. The samples were clarified by centrifugation at 10,000× *g* for 10 min at 4 °C, sterilized using 0.22 µm and 0.45 µm Millex-GP Filter Unit (Merck kGaA), and coated onto wells in 96-well polystyrene microtiter plates (Sumitomo Bakelite, Tokyo, Japan) for biofilm formation assays. This study was approved by the ethics committee of National Institute of Infectious Diseases for usage human saliva in coating plate (IRB approval number: 801). Prior to enrolment, written informed consent was obtained from each subject based on the code of ethics of the World Medical Association (declaration of Helsinki).

### 2.5. Biofilm Formation

Biofilms from each strain were developed in 96-well polystyrene microtiter plates (Sumitomo Bakelite), which were previously coated with human saliva. Biofilm formation assays were performed using a previous procedure at 4 °C overnight [29]. Overnight cultures of *S. mutans* UA159.*gtfBC*^−^ and various bacteria in BHI medium were inoculated at a ratio of 1:100 in 200 µL of TSB with 0.25% sucrose with and without various concentrations of eDNA, MVs, and culture supernatants, from UA159. The plates were incubated at 37 °C in 5% CO_2_ aerobic conditions for 16 h. In the biofilm formation using DNA, *S. mutans* UA159.*gtfBC*^−^ was inoculated at a ratio of 1:100 in 200 µL of TSB, THB, and BHI with 0.25% glucose or 0.25% sucrose with and without various concentrations of various types of DNA from UA159. After incubation, the planktonic cells were removed by washing with distilled water, and the adherent cells were stained as a biofilm formation level with 0.25% safranin for 15 min. After washing with distilled water two times, safranin was extracted from biofilm with 70% (*v/v*) ethanol. Biofilms were quantified by measuring the absorbance of the stained biofilms at 492 nm.

### 2.6. SDS-PAGE and Native-PAGE

Prior to electrophoretic analysis, MVs and the culture supernatants were diluted with an equal volume of native-PAGE or SDS-PAGE buffer (0.06 M of Tris-HCl (Amersham Pharmacia Biotech, Buckinghamshire, UK), pH 6.8, 20% glycerol (Wako Pure Chemical Industries Ltd., Osaka, Japan), 0% or 1% (*wt/vol*) SDS (Wako), 0% or 1% 2-mercaptoethanol (2-ME, Merck kGaA), and 0.0012% bromophenol blue (Wako)). The SDS-PAGE samples were heated at 100 °C for 5 min just prior to loading on the gel. The native-PAGE or SDS-PAGE samples were then ran on a 12.5% polyacrylamide gel (e-PAGEL, ATTO Corp., Tokyo, Japan) in 0.025 M of Tris-HCl, 192 mM of glycine (Wako) and 0% or 0.1% (*wt/vol*) SDS. Electrophoretic separation of the proteins was carried out for 70 min at 25 mA. Gels were stained with Coomassie Brilliant Blue (CBB) and ethidium bromide to observe the proteins and DNA, respectively.

### 2.7. Western Blotting

DNA with proteins were lysed by adding an equal volume of 1 × SDS sample buffer (2% SDS, 50 mM Tris-HCl (pH 6.8), 10% glycerol, 2.5% 2-ME, and 0.1% bromophenol blue) and boiling for 5 min at 95 °C. Equal amounts of each protein sample were then separated by 12.5% acrylamide SDS-PAGE. Subsequently, proteins were transferred onto Immobilon PVDF membranes (Millipore, Bedford, MA) and blocked with 2.5% skim milk in TBST (50 mM Tris, 2.7 mM KCl, 0.138 M NaCl, 0.05% Tween 20, pH 7.6) for 1 h at room temperature. The membranes were probed with anti-GTF antisera [30] from rabbit diluted overnight at 4 °C or with a control non-immunized antibody (Wako) diluted 1:15,000 in 1.25% skim milk/TBST for 1 h at room temperature. HRP-labeled anti-goat secondary antibody (Merck kGaA) was used to detect the antibodies. Optical emission signals on the proteins were produced by enhanced chemiluminescence (ECL Western Blotting Substrate, Thermo Scientific, Southfield, MI, USA) and detected by different time exposures to X-ray film (FUJI FILM, Kanagawa, Japan).

### 2.8. Observation of Live/Dead Cells and Glucan in Biofilm Formation

The biofilm was stained by the FilmTracer Live/Dead Biofilm Viability Kit (Molecular Probes, Inc., Eugene, OR, USA) where SYTO 9 and propidium iodide were added to the biofilms at final concentrations of 5 µM and 30 µM, respectively. The polysaccharides were labeled with Alexa Fluor 647-dextran conjugate (Molecular Probes, Invitrogen Corp., Carlsbad, CA, USA) [31] for red fluorescence, while the nucleic acids in the bacterial cells were labeled with SYTO 9 to produce green fluorescence.

Biofilms were incubated with the dyes at room temperature for 20–40 min before being imaged by a confocal microscope, LSM700 Meta NLO CLSM (Carl Zeiss Inc., Thornwood, NY, USA). Confocal images for the biofilm formation were visually observed by ZEN (Carl Zeiss) analysis software.

### 2.9. Statistical Analyses

The biofilm formation levels were expressed as the mean ± standard deviation. In the biofilm assay, the statistical significance of differences between the bacteria with and without various concentrations of MVs or eDNA were determined using one-way analysis of variance (ANOVA) and Bonferroni correction (IBM SPSS statistics 24, IBM corporation, Armonk, NY, USA). *p*-values less than 0.05 were considered statistically significant. All experiments were repeated independently three times.

## 3. Results

### 3.1. Effects of Various Types of DNA on Biofilm Formation

Short DNA fragments are associated with MVs and may support biofilm formation by MVs. To confirm the role of short DNA fragments in biofilm formation, whole-size pure and short-size pure DNAs were extracted from *S. mutans* UA159. Furthermore, subpurified DNA samples were also extracted as DNA with proteins. At first, we observed DNA and proteins by SDS-PAGE with ethidium bromide and CBB staining in pure DNAs and two types of subpurified DNAs. In the SDS-PAGE and CBB staining, various protein bands were observed in subpurified DNA samples after cell lysis by bead destruction for total 90 and 180 s but not in whole-size pure DNA and subpurified DNA (short-size pure DNA) treated with phenol-chloroform and proteinase K (Figure 1A, center). There were fewer protein bands in subpurified DNA treated with phenol and chloroform than the subpurified DNA without treatments.

In SDS-PAGE and ethidium bromide staining, various short DNA fragments between the 17 kDa and 22 kDa protein markers were observed in all subpurified DNA samples (Figure 1A left). In contrast, the whole-size DNA was observed as a pure DNA. In electrophoresis using an agarose gel and ethidium bromide staining, large and small DNA fragments were observed in whole-size pure DNA and the four subpurified DNA samples, respectively (Figure 1A right). Therefore, sub-purified DNA after beads destruction for total 90 and 180 s indicated short-size fragments with different-size proteins, and subpurified DNA for 180 s treated with phenol and chloroform and proteinase K indicated short-size pure DNA without proteins.

### 3.2. Effects of DNA with Proteins and Whole-Size Pure DNA on Biofilm Formation

The effects of DNA with proteins in sub-purified DNA for 180 s destruction and whole-size pure DNA on biofilm formation were examined using *S. mutans* UA159.*gtfBC*^−^ in TSB with 0.25% sucrose. Low concentrations (0.05 ng/mL) of DNA with proteins induced significant biofilm formation, and the higher biofilm levels continued up to 12.5 ng/mL (Figure 1B). However, whole-size pure DNA did not induce biofilm formation at 0.05–3.1 ng/mL and induced significant biofilm formation at more than 6.3 ng/mL DNA (Figure 1B). There is a possibility that DNA with proteins contains GTF because it induced the biofilm formation of *S. mutans* UA159.*gtfBC*^−^.

To confirm the effects of glucan on the biofilm formation induced by DNA with proteins, observation of glucan synthesis was performed, and the significant production of glucan (red color area) was identified in the biofilm formation induced by DNA with proteins (bead destruction for 90 and 180 s; Figure 2). Moreover, to confirm GTF activities in DNA with proteins in biofilm formation in sucrose conditions (substrate to produce glucan, TSB with 0.25% and 0.5% sucrose, and BHI with 0.25% sucrose) and glucose (non-substrate to produce glucan, TSB and THB with 0.25% glucose), biofilm formations were observed. DNA with proteins significantly induced biofilm formation in sucrose conditions but not in glucose conditions (Figure 3).

There is a possibility that DNA with proteins is linked with the presence of GTF. GTF attached to MVs and formed a complex with short DNA fragments. Therefore, observation of the presence of MVs on DNA with proteins was performed by a scanning electron microscope. Some MVs were found in DNA with proteins (Figure 4A). To confirm the presence of GTF in subpurified DNA, western blotting using anti-GTF serum was performed. Anti-GTF antiserum reacted to 156 kDa bands in sub-purified DNAs (Figure 4B). Therefore, short DNA fragments were produced by bead destruction and bound to GTF attached to MVs.

To compare the biofilm formation abilities between short DNA fragments with and without proteins in detail, these DNA samples were added to the biofilm formation assay using *S. mutans* UA159.*gtfBC*^−^. Short-size pure DNA induced significant biofilms at 0.6 ng/mL but not at 0.15 and 0.3 ng/mL, concentrations at which subpurified DNA indicated the significant biofilm formation but not in full-size pure DNA (Figure 5A). Therefore, short-size pure DNA was more susceptible to significant biofilm formation than full-size pure DNA but lower than sub-purified DNA with proteins. To elucidate the mechanisms for biofilm formation, live/dead cell staining was performed. The biofilms formed by sub-purified DNA with proteins were principally constructed by live cells at all concentrations (Figure 5B). In contrast, the biofilms formed by short-size pure DNA were constructed by live and dead cells at 0.6 ng/ mL (Figure 5C). This indicated that many short-size pure DNAs were physically attached to *S. mutans* UA159.*gtfBC*^−^ and induced attachment and biofilm formation of *S. mutans* UA159.*gtfBC*^−^ on the well surface. Therefore, characteristics of biofilms induced by sub-purified DNA with proteins were different from those induced by short-size pure DNA fragments. The characteristics of biofilms induced by sub-purified DNA with proteins were similar to the GTF-dependent biofilm formation of *S. mutans* in sucrose conditions. We hypothesized that MVs attached to GTF, which is shown in Figure 4, were associated with biofilm formation by sub-purified DNA with proteins.

### 3.3. Effects of GTF-I and GTF-SI on MVs on the Biofilm Formation of S. Mutans

To explore the protein components of the MVs, SDS-PAGE and CBB staining were performed for MVs from *S. mutans* UA159 and GS-5 laboratory strains. One hundred fifty-six kDa and 90 kDa protein bands were observed as the main proteins, and various small bands were also observed (Figure 6A). Culture supernatant without MVs from *S. mutans* GS-5 was removed by ultracentrifugation, and >10 kDa proteins (GSH) were concentrated by centrifugal filters and analyzed using SDS-PAGE. The proteins mainly showed smears at the 15, 50, 70, and 156 kDa bands, but the <10 kDa sample (GSL) did not show clear bands. GTF-I and GTF-SI are the main components for insoluble glucan synthesis related to biofilm formation in *S. mutans.* To confirm the presence of GTF-I and GTF-SI on MVs, MVs from *S. mutans* UA159.*gtfBC*^−^ were extracted and compared with MVs from *S. mutans* UA159 and GS-5 in SDS-PAGE. The 156 kDa bands disappeared; in contrast, the 90 kDa band increased in *S. mutans* UA159.*gtfBC*^−^ (Figure 6A). To identify GTF in the MV samples, Western blotting using anti-GTF antiserum from rabbit, which reacted to the 156 kDa band in both GTF-I and GTF-SI [31], was performed. Strong signals corresponding to the 156 kDa protein band were detected in MVs from UA159 and GS-5, and a weak signal corresponding to the 75 kDa band was detected in the culture supernatant without MVs, >10 kDa proteins but not in UA159.*gtfBC*^−^ (Figure 6B). However, clear signals were not detected in the culture supernatant without MVs in the <10 kDa proteins (Figure 6B). We confirmed that MVs included GTF as the main protein. To study the activities of MVs in biofilm formation, MVs from UA159, UA159.*gtfBC*^−^, GS-5, and culture supernatant without MVs samples were added to the biofilm formation assay using UA159.*gtfBC*^−^ in TSB with 0.25% sucrose. MVs from UA159 and GS-5 induced significantly higher biofilms at more than 0.1 μg/mL than no sample (*p* < 0.05), but the culture supernatant without MVs samples did not (Figure 6C). The biofilm formation slowly increased in a dose-dependent manner and reached a peak at 10 μg/mL, but its level was lower in the culture supernatant without MVs with >10 kDa proteins than in biofilm levels with 0.1 μg/mL MVs (Figure 6C). The biofilm formation by culture supernatant sample without MVs from UA159 was also similar to those from GS-5 (data not shown). MVs from UA159.*gtfBC*^−^ did not show significant biofilm formation (Figure 6D).

To remove deposits that were weakly and accidentally attached with MVs, MVs (MVW) were washed with sterile PBS in ultracentrifugation. To remove eDNA bound to MVs, the washed MVs (MVWD) were pretreated by 100 units/mL DNase I at 37 °C for 1 h and washed with sterile PBS to remove free DNase I by ultracentrifugation again. After washing with PBS and treatment of DNase I, the main protein at 156 kDa of GTF was maintained (Appendix A). To confirm the presence of eDNA in MV samples, the SDS-PAGE gel was stained by ethidium bromide. Bands in the bottom area at less than 10 kDa through 12.5% acrylamide gel fluoresced in MV, MVW, and MVWD (Appendix A). It was found that the MVs bound to short size DNA and RNA. To observe the native protein status of MVs, native-PAGE was performed. All proteins did not show clear bands and showed smeared bands (Appendix A). In particular, the smeared bands disappeared after washing in MVW. When stained with ethidium bromide, the DNA or RNA in MV and MVW that remained near the stacking gel groove fluoresced (Appendix A). It was confirmed that DNA and RNA were almost attached to MVs that remained in the gel groove. Concentrations of dsDNA and RNA were quantitatively measured by a NanoDrop^TM^ Lite UV-Vis spectrophotometer in MV samples. High concentrations of dsDNA and RNA were observed in independent MV1 and MV2 concentrated by centrifugal filters for >10 kDa and >50 kDa, respectively (Appendix A). In the MV2W and MV2WD, the concentrations of dsDNA and RNA decreased considerably. MVs made complexes with dsDNA and RNA. The biofilm formation activities of MVW or MVWD to *S. mutans* UA159.*gtfBC*^−^ and *Actinomyces naeslundii* × 600 were observed and compared with those in MF and GSsup >50 (Appendix A). The biofilm formation activities of MVW and MVWD were similar to those of MV at more than 0.63 μg/mL. These results indicated that complex DNA, RNA and MVs attached with GTF had activities to biofilm formation of *S. mutans* and *A. naeslundii*.

To determine the effects of MVs on the biofilm formation of various microorganisms in oral biofilm bacteria, other microorganisms including *C. albicans* were inoculated into the biofilm formation assay involving MVs from UA159 or UA159*.gtfBC*^−^ in TSB with 0.25% sucrose. MVs from UA159 up-regulated the biofilm formation of initial colonizers on the tooth surface [32], such as *S. mitis*, *S. oralis*, *S. sanguinis*, *S. gordonii*, *A. naeslundii*, and *Actinomyces oris*, but MVs from UA159.*gtfBC*^−^ did not up-regulated biofilm formation excepting for *A. naeslundii* (Table 1). MVs from UA159 did not stimulate other bacteria, such as *S. salivarius* and *Neisseria* sp., or the *Streptococcus milleri* group, such as *S. anginosus*, *S. intermedius*, *S. pyogenes*, *C. albicans*, and *S. aureus.* Therefore, MVs specifically induced the biofilm formation by multiple oral bacterial species including initial colonizers depending on GTF-I and GTF-SI.

The data indicate the mean ± standard deviation (SD) of triplicate experiments in the biofilm formation assessed by absorbance at 492 nm. The independent experiments were performed three times, with similar results obtained in each. The asterisks indicated a significant difference between the two groups (ANOVA with Bonferroni correction; *: *p* < 0.05, UA159 MVs vs. no sample or UA159.*gtfBC*^−^ MVs). UP and DOWN: The biofilm formation was significantly enhanced and reduced by addition of UA159 MVs.

## 4. Discussion

Insoluble glucans synthesized by GTF-I and GTF-SI is the principal components for the development of pathogenic biofilms associated with dental caries [8,10,33,34]. Moreover, eDNA interaction with GTF-I and GTF-SI has important roles in glucan production, *S. mutans* colonization and biofilm matrix assembly [18,35]. eDNA has been found in high amounts in the matrix of biofilms involving insoluble glucan [19]. eDNA is a byproduct of autolysis [36,37] and is secreted with MVs [19]. Synthesis of peptidoglycan affects production of GTFs and the release of eDNA and MVs [38]. MVs produced from the cytoplasmic membrane are linked to their release in physiological formation or cell lysis during cell growth and are associated with GTF-I and GTF-SI and DNA release.

Extra cellular DNA is one of the contributors to biofilm formation and has important functions in the attachment and an aggregation of bacteria on the surface for initial stage of the biofilm formation [15,25]. In this study, sub-purified DNA with proteins induced significant biofilm formation of *S. mutans* UA159*.gtfBC*^−^ live cells in conditions without insoluble glucan and with soluble glucan and fructan. Previous reports showed that the presence of fructan and eDNA and the absence of insoluble glucan induced significant *S. mutans* biofilm formation on human saliva-coated hydroxyapatite disks in raffinose-supplemented conditions [26]. Therefore, soluble polysaccharides, such as fructan, are required for eDNA-dependent biofilm formation in conditions without insoluble glucan. However, an opportunity and function for producing DNA is unknown and actually is required in the attachment of aggregated single cells on the surfaces during the initial stage. Moreover, this report suggested that the hydrophobicity on the cell surface of *S. mutans* is important for the production of eDNA in the conditions supplemented with a small amount (0.0075% (*w/v*)) of sucrose [26] and 0.25% raffinose. Small amounts of insoluble glucan, which are synthesized in the condition, may contribute to the up-regulation of hydrophobicity on the cell surface and induce cell aggregation and accumulation. In a previous report, it was suggested that dietary sucrose and starch enhance release of eDNA into the matrix during *S. mutans* biofilm formation [35]. In addition, eDNA increased glucan synthesis by GtfB adsorbed on the saliva-coated hydroxyapatite and *S. mutans* surface [33]. These reports indicate the interaction between eDNA and glucan resulting in elements for biofilm formation systems such as aggregation and adherence. The origin of DNA in biofilms could be from the genomic DNA due to the cell lysis of the subpopulation by active secretion [39,40,41]. Therefore, bacterial eDNA is naturally produced and plays a role in appropriate conditions for aggregating bacteria to each other and the adherence of cells on the surface.

This present study indicated that extracellular MVs with GTFs were associated with short size DNA and contributed the biofilm formation of other bacterial species. This complex strongly induced the GTF-dependent biofilm formation of not only *S. mutans* UA159.*gtfBC*^−^ but also early colonizers on the tooth surface, such as *S. mitis*, *S. oralis*, *S. sanguinis*, *S. gordonii*, and *Actinomyces* spp. [32]. In a previous report, eDNA increased glucan synthesis by GTF-I adsorption on saliva-coated hydroxyapatite and the *S. mutans* surface [33]. Therefore, this formation of MVs and eDNA may be helpful for the effects of GTF-I and GTF-SI on biofilm formation by bacterial species, which were present as *S. mutans*-independent oral biofilm bacteria in a previous paper [32]. The cargo of the vesicle reflects the physical state of the cell at the time of death rather than being the consequence of a specific sorting mechanism [22]. Rainey et al. indicated that changes in gene expression in acidic pH might lead to activated secretion of MVs and eDNA [41]. Kawarai et al. showed that changes from neutral pH primary conditions to slightly low pH conditions caused glucan-independent and eDNA-dependent biofilm formation of *S. mutans* in conditions with sucrose [26]. eDNA has multiple functions due to the involvement of acid-base interaction in the attachment and aggregation of bacteria on the surface for the initial stage of biofilm formation [15,25]. *S. mutans*, which is a late colonizer on the tooth surface, produces eDNA after long-recognized primary colonizers, such as *S. mitis*, *S. oralis*, and *S. sanguinis*, and reinforces their attachment and colonization.

Investigators reported that the major components of the biofilm matrix of *Staphylococcus aureus* are proteins and eDNA that depend on a drop in pH during growth in the presence of glucose [42,43]. Other investigators suggested that some extracellular DNA and amyloid-like complexes closely interacted within the biofilm matrix [44]. Biofilm formation depended on the presence of both DNA and proteins in sub-purified DNA but not in purified whole size DNA without proteins. It was clarified that various proteins attached to short size DNA are important for eDNA-dependent biofilm formation. Pure short DNA fragments, in which proteins and MVs were removed by proteinase K, induced higher biofilm formation than full-size DNA at an appropriate concentration. Affinities of short DNA fragments to bacterial cells may have arisen because of the shorter size and removal of proteins. Shorter size and pure DNA have physical benefits for penetration because of the depressions on the MVs with GTFs. Oligonucleotides are strong polyelectrolytes carrying a univalent negative charge per base at pH > 2 that contain nitrogenous bases that are hydrophobic in nature [45]. Both electrostatic and hydrophobic interactions are major driving forces for short chain 20-mer pyrimidinic homo-ss-DNA adsorption at solid–liquid interfaces [46]. When MVs are produced during cell growth, short eDNA fragments may interact with MVs with GTFs under the electrostatic and hydrophobic interactions. Cell lysis by bead destruction for 90 and 180 s induced short DNA fragments that attached to MVs and GTF. This suggests that cell lysis leads to the production of the complex of MVs, DNA, and GTF in natural growth.

Recently, Koo et al. provided a rational for multi-targeted therapies that are aimed at disrupting the complex biofilm microenvironment involving in *S. mutans* [47]. Extra cellular polysaccharides such as glucan, GTF, glucan binding protein, quorum sensing, and eDNA were picked up and summarized as target. Short size DNAs and complex form of short size DNA associated with GTF and MVs presented in this study are important factors for the formation of biofilms by oral multi-species microorganisms and may be one of additional targets for the prevention of oral biofilm-associated diseases. However, our study does not fully elucidate the production mechanisms of short size DNA and MVs. Further study of the mechanism of decreasing from whole size to short size DNA, the bonding of DNA and GTF on the MVs and production of MVs are necessary to understand the eDNA and MVs-dependent biofilm formation.

## 5. Conclusions

Cell lysis of *Streptococcus mutans* by bead destruction for 90 and 180 s induced short DNA fragments that attached to membrane vesicles (MVs) and glucosyltransferases (GTFs). This suggests that cell lysis leads to the production of the complex of MVs, DNA, and GTFs in natural growth. Pure short DNA fragments, in which proteins and MVs were removed by proteinase K, induced higher biofilm formation of *S. mutans* than full-size DNA. The complex of MVs, DNA, and GTFs induced the GTFs-dependent biofilm formation of early colonizers on the tooth surface. The short size DNAs associated with MVs and GTFs are important contributors for the formation of biofilms by oral pathogen and may be one of additional targets for the prevention of oral biofilm-associated diseases.

## Figures and Tables

**Figure 1 molecules-24-03131-f001:**
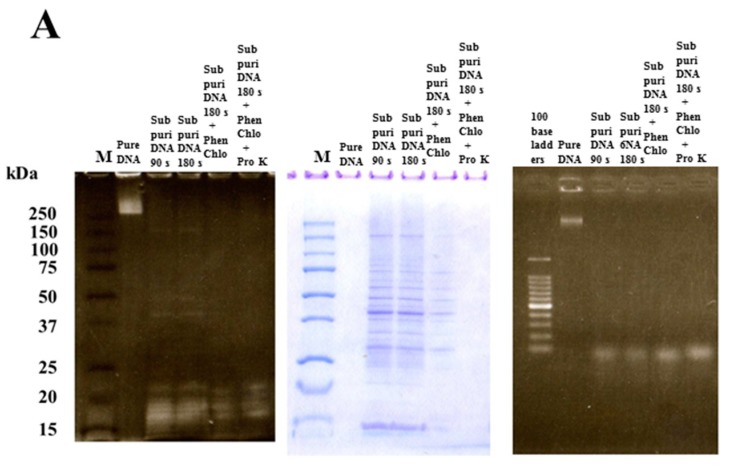
Preparation of various types of DNA. (**A**) Whole-size pure DNA, sub-purified DNAs prepared by beads destruction for 90 (Sub puri DNA 90s) and 180 s (Sub puri DNA 180 s), sub puri DNA 180 s treated with phenol and chloroform (Sub puri DNA 180 s + Phen Chlo), and sub puri DNA 180 s + Phen Chlo treated with proteinase K (sub puri DNA 180 s + Phen Chlo + Pro K) were observed for molecular size and proteins. Left: Polyacrylamide gel after SDS-PAGE analysis using various types of DNA was stained by Coomassie Brilliant Blue (CBB), and Center: Ethidium bromide staining to left gel. Right: Agarose gel after electrophoresis using the same samples was stained by ethidium bromide. M: Protein marker. (**B**) Sub-purified DNAs prepared by beads destruction for 180 (Sub puri DNA 180 s) was used as a DNA with proteins. Biofilm formation of *Streptococcus mutans* UA159 *gtfBC^−^* was quantitatively assessed in conditions with DNA with proteins and pure full size DNA. The data indicate the mean ± standard deviation (SD) of triplicate experiments. The independent experiments were performed three times, with similar results obtained in each. The asterisks indicate a significant difference between the two groups (*: *p* < 0.01, no DNA vs. DNA).

**Figure 2 molecules-24-03131-f002:**
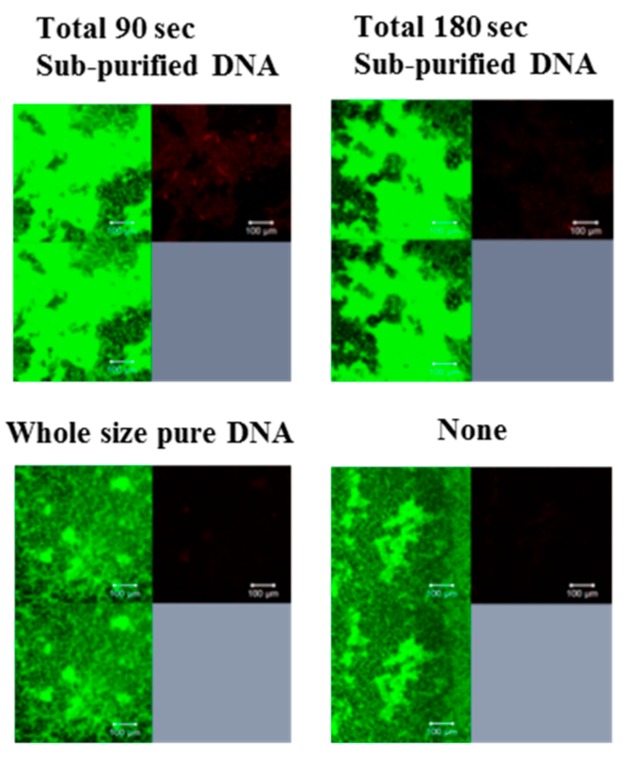
Roles of glucan for biofilm formation of *S. mutans* by sub-purified DNA. The biofilm formation of *S. mutans* UA159 *gtfBC^−^* induced by sub-purified DNA were analyzed for the production of glucan and observed in media including sucrose. The biofilm formations of *S. mutans* UA159 *gtfBC^−^* were induced by none, sub-purified DNA prepared by beads destruction for total 90 and 180 s, and whole size pure DNA. Their biofilms were stained by SYTO 9 and Alexa Flour 647-dextran conjugate, observed by confocal microscope and analyzed by Zen. Upper left: Live cells, Upper right: Glucan, and Lower left: Live cell merged with glucan were presented. Representative data from more than three independent experiments are presented in the pictures.

**Figure 3 molecules-24-03131-f003:**
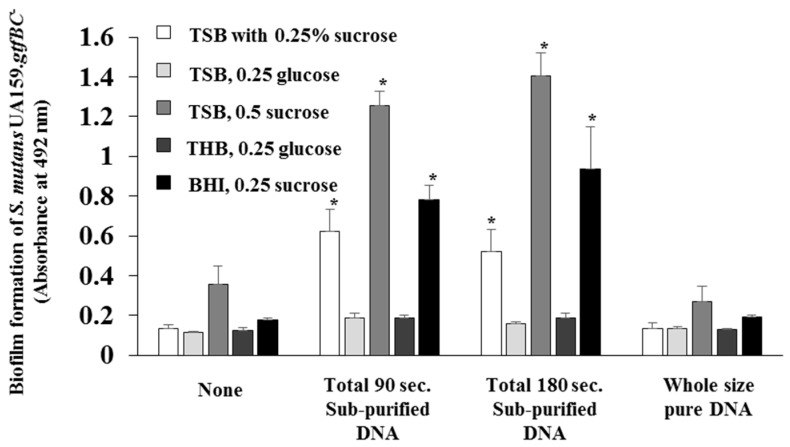
Effects of carbohydrates on the biofilm formation by DNA samples. Biofilm formations were quantitatively assessed in none, sub-purified DNA prepared by beads destruction for a total of 90 and 180 s, and whole-size pure DNA in TSB with 0.25% sucrose, 0.25% glucose, and 0.5% sucrose, THB with 0.25% glucose, and BHI with 0.25% sucrose. The data indicate the mean ± standard deviation (SD) of triplicate experiments. The independent experiments were performed three times, with similar results obtained in each. The asterisks indicate a significant difference between the two groups (*: *p* < 0.05, no DNA vs. DNA).

**Figure 4 molecules-24-03131-f004:**
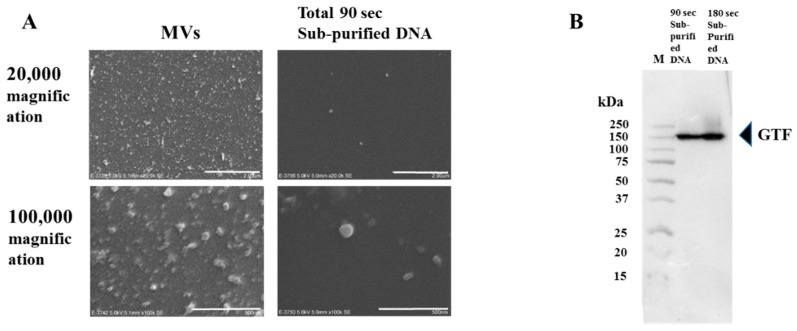
Presence of GTF-I and GTF-SI in sub-purified DNA. (**A**) Membrane vesicles (MVs) and sub-purified DNA prepared by beads destruction for a total of 90 s from *S. mutans* UA159 were observed at 20,000 and 100,000 magnifications by electron microscope. The independent experiments were performed three times, with similar results obtained in each. (**B**) Sub-purified DNAs prepared by beads destruction for 90 and 180 s in *S. mutans* UA159 were applied into SDS-PAGE and western blotting using anti-GTF antiserum was performed. Representative data from more than three independent experiments were presented in the pictures.

**Figure 5 molecules-24-03131-f005:**
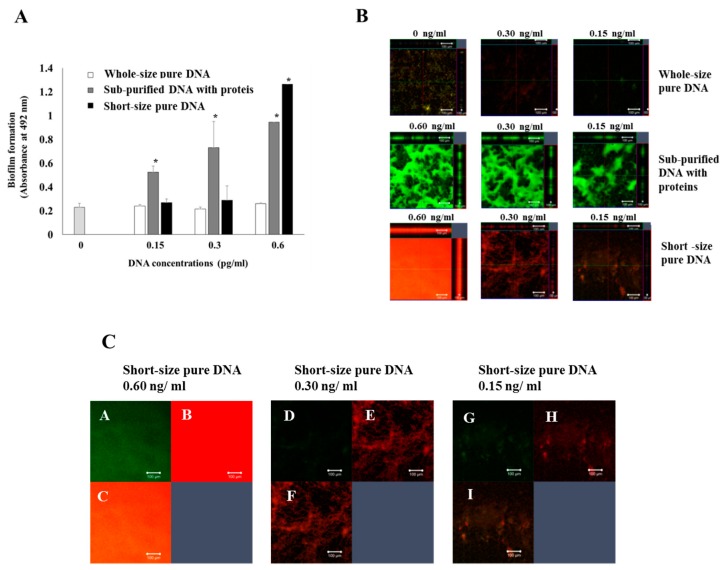
Biofilm formation of *S. mutans* by DNA. Biofilm formations induced by 0, 0.15, 0.3, and 0.6 ng/mL of whole-size pure DNA (open bar), sub-purified DNA with proteins (gray bar) from 180-s cell destruction and short-size pure DNA in UA159 were quantitatively assessed by absorbance at 492 nm (**A**). The data indicate the mean ± standard deviation (SD) of triplicate experiments. The independent experiments were performed three times, with similar results obtained each time. The asterisks indicate a significant difference between the two groups (*: *p* < 0.05, no DNA vs. DNA). Biofilm formations induced by 0, 0.15, 0.3, and 0.6 ng/mL of whole-size pure DNA, sub-purified DNA with proteins and short-size pure DNA were visually observed by confocal microscope with live/dead staining (**B**). *X*–*Y*, *X*–*Z*, and *Y*–*Z* axes biofilms are shown in each picture. Representative data from more than three independent experiments are shown in the pictures. The biofilm formations of *S. mutans* UA159*.gtfBC*^−^, which were induced by various concentrations of short-size pure DNA, were stained by the Live/Dead BacLight Viability Kit, observed by confocal microscope and analyzed (Zen). Live (A, D, G), dead (B, E, H), and merged cells (C, F, I) in quadrant square images are shown (**C**). Representative data from more than three independent experiments are shown in the pictures.

**Figure 6 molecules-24-03131-f006:**
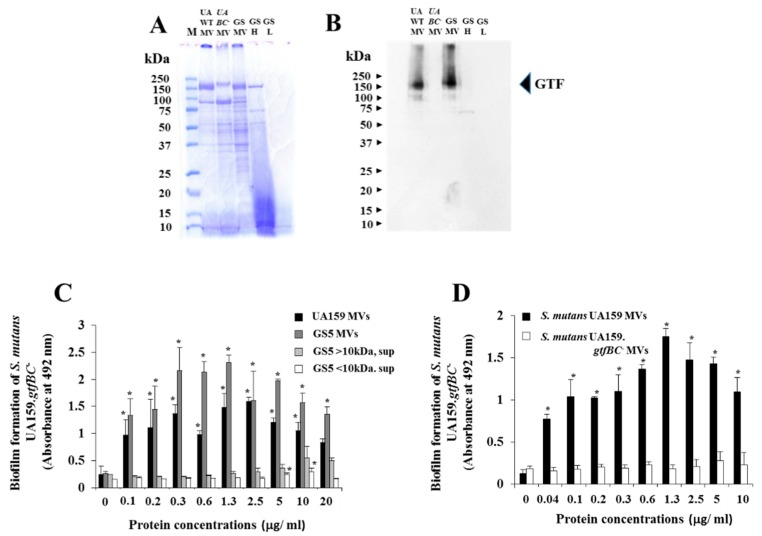
Biofilm formation of *S. mutans* by MVs. (**A**) MVs from *S. mutans* UA159 (UAWTMV), UA159*.gtfBC*^−^ (UABC^-^MV), GS-5(GSMV), and culture supernatant without MVs and >10 kDa (GSH) and <50 kDa (GSL) proteins from GS-5 were analyzed by SDS-PAGE, and the gel was stained by CBB. M: Protein markers. (**B**) Western blotting using anti-GTF antisera was performed after SDS-PAGE. (**C**) Biofilm formation of *S. mutans* UA159*.gtfBC*^−^ was observed in conditions with added MVs from UA159 and culture supernatant without MVs and >10 kDa and <10 kDa proteins from GS-5. (**D**) Biofilm formation of *S. mutans* UA159*.gtfBC*^−^ was observed in conditions with MVs from UA159 and *S. mutans* UA159*.gtfBC*^−^. The data indicate the mean ± standard deviation (SD) of triplicate experiments. The independent experiments were performed three times, with similar results obtained each time. The asterisks indicate a significant difference between the two groups (*: *p* < 0.05, no MVs vs. MVs).

**Table 1 molecules-24-03131-t001:** GTF-I and GTF-SI on biofilm formation in oral microorganisms.

Strains		None	UA159 MVs	UA159*.gtfBC*^−^ MVs	Activity
*S. mitis*	ATCC 903	0.241 ± 0.027	1.370 ± 0.101 *	0.242 ± 0.048	UP
	ATCC 6249	0.411 ± 0.045	0.432 ± 0.082	0.473 ± 0.082	
*S. oralis*	ATCC 35037	0.149 ± 0.012	0.685 ± 0.079 *	0.127 ± 0.002	UP
*S. sanguinis*	ATCC 10556	1.147 ± 0.256	2.029 ± 0.082 *	1.430 ± 0.240	UP
	ST 205	0.270 ± 0.035	1.164 ± 0.221 *	0.356 ± 0.044	UP
	ST 134	0.175 ± 0.006	0.255 ± 0.044	0.160 ± 0.030	
*S. gordonii*	ATCC 10558	0.406 ± 0.018	1.727 ± 0.341 *	0.398 ± 0.005	UP
*S. anginosus*	ATCC 33397	1.256 ± 0.254	1.175 ± 0.026	1.115 ± 0.581	-
*S. intermedius*	ATCC 27335	0.153 ± 0.010	0.330 ± 0.070	0.150 ± 0.036	-
*S. pyogenes*	K33	0.289 ± 0.011	0.377 ± 0.182	0.358 ± 0.163	-
	K32	0.574 ± 0.107	0.484 ± 0.091	0.529 ± 0.330	-
*S. pneumoniae*	GTC 261	0.178 ± 0.015	0.127 ± 0.010	0.212 ± 0.133	-
*S. salivarius*	JCM 5907	0.492 ± 0.018	0.584 ± 0.020	0.566 ± 0.012	-
	ATCC 9759	0.124 ± 0.021	0.299 ± 0.070	0.163 ± 0.020	-
	HT9R	0.222 ± 0.029	0.277 ± 0.005	0.244 ± 0.007	-
*A. naeslundii*	x600	0.415 ± 0.092	1.916 ± 0.582 *	1.059 ± 0.423	UP
*A. oris*	MG1	1.087 ± 0.222	2.264 ± 0.207 *	0.998 ± 0.157	UP
*N. cinerea*	23-1	0.344 ± 0.018	0.261 ± 0.011	0.339 ± 0.011	-
*N. flavescens*	ATCC 13120	0.353 ± 0.012	0.231 ± 0.044	0.376 ± 0.010	-
*N. mucosa*	16-2	1.889 ± 0.260	1.333 ± 0.133	1.590 ± 0.172	DOWN
*N. subflava*	#2	1.071 ± 0.101	1.318 ± 0.237	1.301 ± 0.208	-
*C. albicans*	SC5314	0.393 ± 0.046	0.292 ± 0.063	0.433 ± 0.181	-
	SC5312	0.178 ± 0.029	0.288 ± 0.049	0.171 ± 0.015	-
*S. aureus*	Cowan I	0.283 ± 0.004	0.328 ± 0.018	0.262 ± 0.043	-
	ATCC 6538P	0.211 ± 0.036	0.269 ± 0.036	0.208 ± 0.010	-

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
