# Peer review of "Effects of Complex DNA and MVs with GTF Extracted from Streptococcus mutans on the Oral Biofilm"

_molecules, 2019, doi:10.3390/molecules24173131_

Round 1

Reviewer 1 Report

The study assesses whether bacterial biomass differs among biofilm colonies across a range of extraction protocols. The aim of the study seems to be an assessment of the size fractioning of the different levels. There is a great deal of detail regarding the extraction and culture methods, however the study design and statistical rational need to be clarified and reassessed. The connection with eDNA is not clear as the study utilizes a set of cultures to perform their test without directly testing the findings against the eDNA source being alluded to (e.g. plaque). I find the over use of acronyms make reading the manuscript difficult to follow as well and would suggest writing these out. The biomass estimates are not biomass calculations, but rely on a single light reflectance measure from a single wavelength, which is known to have large random variation in readings from samples and machine operation. There is also a general concern for the application of multiple test to drawn general. Overall, the study needs to be refined/rewritten to address a clear question along with statistical considerations for the repeated measures, which are not addressed in the current version.

Abstract: include sample size the experimental groups and clarify the exploratory groups (e.g. ‘DNA size: three levels’). Also include the statistics for the statements where significant differences are implied and explain how the levels are different (e.g. ‘short-size DNA showed significantly higher biofilm accumulation compared to xxx (p = xxx, sd = xxx)’).

56-58- Environmental DNA is DNA obtained from environmental samples (here plaque, but can be from soil, water, pollen etc.) that is taken without targeting a particular species or taxon group.

Deiner, K., Bik, H.M., Mächler, E., Seymour, M., Lacoursière‐Roussel, A., Altermatt, F., Creer, S., Bista, I., Lodge, D.M., De Vere, N. and Pfrender, M.E., 2017. Environmental DNA metabarcoding: Transforming how we survey animal and plant communities. Molecular ecology, 26(21), pp.5872-5895.

Torti, A., Lever, M.A. and Jørgensen, B.B., 2015. Origin, dynamics, and implications of extracellular DNA pools in marine sediments. Marine genomics, 24, pp.185-196.

63-67: Define what is meant by ‘pure DNA’ and ‘sub-purified DNAs’ – protein-free fragments are alluded to, but it is unclear what the relevant variation in factor levels are which makes it difficult to assess the balance of the study design.

75-87: clearly state the number of utilized strains and the number of utilized species.

90: Clearly state the number of DNA extractions performed.

90-122: cryptic and unclear regarding what steps correspond to different factor levels (i.e. DNA sizes). the second and third factor levels are mixed making the separation unclear and are left without clear definitions. The number of factor levels (five) does not match the level number implied in the introduction/abstract (three).

140-141: human saliva from a single lab tech after their lunch or from a pool of individuals?

185-189: The experimental design needs to be made much clearer to allow assessment. Biomass measures are not previously explained in the methods. Biomass will need to be standardized across the different extraction methods to account for variation in the methods. The statistics should include a single response variable (e.g biomass) in relation to the fixed/explanatory variables (e.g. extraction type). The current text implies multiple test were performed which, given the study design, would not be appropriate. There should be some consideration for the expected hypotheses with such test. Comparing biomass levels from a purified culture against one that is not purified (for example) is not informative without first stating what the expectations are from previously conducted experiments or the expected underlying mechanisms.

Figure 2: This should be two separate figures. Figure legends should be decipherable without references to the main text, so the acronym usage needs to be minimized and the sample size per level needs to be provided. Splitting the figure into two figures will also reduce the legend text load.

260-261: As the experiment was repeated multiple times, there needs to be a repeated measures covariate included in the statistical analyses. Was there no within experiment replication to assess sd?

301-325: There is not statistical support given for any of the statements provided.

Results: biomass values are missing from the results and need to be provided for all levels of the experiment.

The use of a single light absorbance band for biofilm mass needs to be fully justified. why not use other bands which will give different values? Additionally, the machine employed is known to be highly variable in the output across samples and operations, making the numbers obtained highly suspect and are generally only used for quality and diagnostic purposes.

Author Response

Responses to comments

The study assesses whether bacterial biomass differs among biofilm colonies across a range of extraction protocols. The aim of the study seems to be an assessment of the size fractioning of the different levels. There is a great deal of detail regarding the extraction and culture methods, however the study design and statistical rational need to be clarified and reassessed. The connection with eDNA is not clear as the study utilizes a set of cultures to perform their test without directly testing the findings against the eDNA source being alluded to (e.g. plaque).

Answer) I understood reviewer’s comments about directly testing the findings against the eDNA. There are many bacterial species in plaque. In this study, eDNAs from S. mutans laboratory strains, UA159, GS-5 had function to induce GTF-dependent biofilm formation of S. mutans gtfBC- and initial colonizers on the tooth surface. eDNA from S. mutans clinical isolate FSM-5, FSC-3 showed similar function to UA159 and GS-5. However, eDNA from other species showed similar function to S. mutans gtfBC-, which lost GTFs, but not have same functions as S. mutans. Therefore, eDNA from plaque show mixture effects by eDNAs from S. mutans and other bacterial species on the biofilm formation. This study presented connecting form of short-size eDNA and MVs with GTF in S. mutans. This from had specific function to the biofilm formation by multiple species.

I find the over use of acronyms make reading the manuscript difficult to follow as well and would suggest writing these out.

Answer) I agreed reviewer’s comments about use of acronyms. In Fig. 3, “s” and “g” have been changed to “sucrose” and “glucose”. In other words, acronyms has been corrected and some words have been added to explain acronyms.

The biomass estimates are not biomass calculations, but rely on a single light reflectance measure from a single wavelength, which is known to have large random variation in readings from samples and machine operation.

Answer) I understood reviewer’s comments about single light reflectance measure.

We did not measure biomass in the biofilm in this study. P. 10 line 229. This is miss description. “biomass” has been changed to “the biofilm formation levels”. The biofilm formations were stained with 0.25% safranin and, after washing with distilled water, safranin was extracted from the biofilm with 100 ml of 70% (vol/vol) ethanol, and assessed by absorbance at 492 nm. This assessment levels are reflected to the biofilm formation levels. This methods were referred from our group paper which was published in Journal (29). However, I missed description about assessment of biofilm. P. 8 line 186, a sentence “After washing with distilled water 2 times, safranin was extracted from biofilm with 70 % (vol / vol) ethanol.” has been inserted before assessment of biofilm by 492 nm.

There is also a general concern for the application of multiple test to drawn general. Overall, the study needs to be refined/rewritten to address a clear question along with statistical considerations for the repeated measures, which are not addressed in the current version.

 Answer) I agreed reviewer’s comments about use of statistical analysis. P. 5 line 189, all experiments were repeated independently three times. However, I missed description about independent experiments three times in some figures. P. 12 Fig. 1, p. 14 Fig. 3, p. 16 Fig. 5 and p. 18 Fig, 6, “independent” has been added between “The” and “experiments”. P. 15 Fig. 4, a sentence “The independent experiments were performed three times, with similar results obtained in each” has been added.

Abstract: include sample size the experimental groups and clarify the exploratory groups (e.g. ‘DNA size: three levels’).

 Answer) I agreed reviewer’s comments about use of sample size in Abstract. P. 2 line 30, “short and whole sizes” has been added. P. 2 line 31, “2 types of “ has been added. Line 38, “whole-size pure DNA” has been added. Line 39, “Short-size pure DNA” has been added. Line 41, “short-size DNA associated with MVs” has been added.

Also include the statistics for the statements where significant differences are implied and explain how the levels are different (e.g. ‘short-size DNA showed significantly higher biofilm accumulation compared to xxx (p = xxx, sd = xxx)’).

Answer) I agreed reviewer’s comments about statistics for the statements in Abstarct.

Line 37-38, “than no sample (p < 0.05)” has been added. Line 41, “than no sample (p < 0.05.” has been added. We performed ANCOVA analysis about effects of sucrose on the biofilm formation by DNA. There are no covariate effects within experiment replication in relationship of biofilm formation between presence of sucrose and sub-purified DNA with proteins.

56-58- Environmental DNA is DNA obtained from environmental samples (here plaque, but can be from soil, water, pollen etc.) that is taken without targeting a particular species or taxon group.

 Answer) Thank you for your good suggestion about environmental DNA. P. 4 line 77-78, a sentence “eDNAs are environmentally obtained from plaque in oral cavity that are taken without targeting a particular species” has been added.

63-67: Define what is meant by ‘pure DNA’ and ‘sub-purified DNAs’ – protein-free fragments are alluded to, but it is unclear what the relevant variation in factor levels are which makes it difficult to assess the balance of the study design.

 Answer) Thank you for your suggestion about description of eDNA samples. P. 4 line 85-86, 4 types of DNA (short and whole sizes pure DNAs, and 2 types of sub-purified DNAs) were described.

75-87: clearly state the number of utilized strains and the number of utilized species.

 Answer) I agreed reviewer’s comments about the number of utilized strains and the number of utilized species.

5 line 97, a sentence “Twenty eight in 17 bacterial species and 2 fungus were used for experiments.” has been added.

90: Clearly state the number of DNA extractions performed.

  Answer) I agreed reviewer’s comments about the number of DNA extractions.

5 line 113, “Four types of DNA (whole-size and short size pure DNA, and sub-purified DNAs after total 90 and 180 seconds beads destruction)” has been added.

90-122: cryptic and unclear regarding what steps correspond to different factor levels (i.e. DNA sizes). the second and third factor levels are mixed making the separation unclear and are left without clear definitions. The number of factor levels (five) does not match the level number implied in the introduction/abstract (three).

   Answer) I agreed reviewer’s comments about the steps of DNA extractions.

5 line 116-117, a sentence has been revised. 6 line 121-124, a sentence has been revised.

Line 134, “short-size pure DNA” has been aaded.

Line 142-144, a sentence have been added.

140-141: human saliva from a single lab tech after their lunch or from a pool of individuals?

   Answer) I agreed reviewer’s comments about description of human saliva.

7 line166-P. 8 line 175, a paragraph has been added.

185-189: The experimental design needs to be made much clearer to allow assessment. Biomass measures are not previously explained in the methods. Biomass will need to be standardized across the different extraction methods to account for variation in the methods. The statistics should include a single response variable (e.g biomass) in relation to the fixed/explanatory variables (e.g. extraction type). The current text implies multiple test were performed which, given the study design, would not be appropriate. There should be some consideration for the expected hypotheses with such test. Comparing biomass levels from a purified culture against one that is not purified (for example) is not informative without first stating what the expectations are from previously conducted experiments or the expected underlying mechanisms.

 Answer) Thank you for your suggestion. We did not measure biomass in the biofilm in this study. This is miss description. P. 10 line 231. “biomass” has been changed to “the biofilm formation levels”. The biofilm formations were stained with 0.25% safranin and, after washing with distilled water, safranin was extracted from the biofilm with 100 ml of 70% (vol/vol) ethanol, and assessed by absorbance at 492 nm. This assessment levels are reflected to the biofilm formation levels. This methods were referred from our group paper which was published in Journal (31). However, I missed description about assessment of biofilm. P. 8 line 188-189, a sentence “After washing with distilled water 2 times, safranin was extracted from biofilm with 70 % (vol / vol) ethanol.” has been inserted before assessment of biofilm by 492 nm.

Figure 2: This should be two separate figures. Figure legends should be decipherable without references to the main text, so the acronym usage needs to be minimized and the sample size per level needs to be provided. Splitting the figure into two figures will also reduce the legend text load.

  Answer) Thank you for your suggestion. Figure 2 has been divided for 2 figure (Fig. 2 and Fig. 3). The acronym has changed to full name in Figure 3. Whole-size pure DNA has been described in Figure 2 and Figure 3.

260-261: As the experiment was repeated multiple times, there needs to be a repeated measures covariate included in the statistical analyses. Was there no within experiment replication to assess sd?

 Answer) Thank you for your suggestion about covariate. We performed ANCOVA analysis about effects of sucrose on the biofilm formation by DNA. There are no covariate effects within experiment replication in relationship of biofilm formation between presence of sucrose and sub-purified DNA with proteins.

301-325: There is not statistical support given for any of the statements provided.

Answer) Thank you for your suggestion about statistical support in Fig. 6.

18 line 325-327, a sentence “MVs from UA159 and GS-5 induced significantly higher biofilms at more than 0.1 mg/ ml than no sample (p < 0.05)” has been added.

Results: biomass values are missing from the results and need to be provided for all levels of the experiment.

Answer) Thank you for your suggestion about biomass values. We did not measure biomass in the biofilm in this study.

The use of a single light absorbance band for biofilm mass needs to be fully justified. why not use other bands which will give different values? Additionally, the machine employed is known to be highly variable in the output across samples and operations, making the numbers obtained highly suspect and are generally only used for quality and diagnostic purposes.

Answer) Thank you for your suggestion about a single light absorbance. The biofilm formations were stained with 0.25% safranin and, after washing with distilled water, safranin was extracted from the biofilm with 100 l of 70% (vol/vol) ethanol, and assessed by absorbance at 492 nm. This assessment levels are reflected to the biofilm formation levels. This methods were referred from our group paper which was published in Journal (29). Recently, I have published many biofilm formation manuscripts using S. mutans, Actinomyces naselundii and Actinomyces oris. This absorbance assessment procedure was used above manuscripts. Color of safranin could be assessed by absorbance at 492 nm. Other wavelengths were not matched with color of safranin as absorbance. This method is available for the assessment of biofilm formation level on the plate.

English spell and grammars were checked by a English speaker and revised carefully.

Reviewer 2 Report

The study presents interesting results and new knowledge on the participation of the complex of DNA and MVs with the GTF extracted from Streptococcus mutans in an oral biofilm model. The methodology is correctly delineated and the results are clearly described. However, the ethics committee approval number is not provided. Indicate if human saliva was obtained from people or artificial saliva was used.

Author Response

Responses to comments

The study presents interesting results and new knowledge on the participation of the complex of DNA and MVs with the GTF extracted from Streptococcus mutans in an oral biofilm model. The methodology is correctly delineated and the results are clearly described. However, the ethics committee approval number is not provided. Indicate if human saliva was obtained from people or artificial saliva was used.

Answer) I am glad to hear reviewer’s above comments. I have added “Human saliva collection” in Materials and methods and added the ethics committee approval number.

English spell and grammars were checked by a English speaker and revised carefully.

Reviewer 3 Report

Introduction: Rewrite the sentence: „when various antibacterial agents, 31 such as lactoferrin, lactoperoxidase, and lysozyme, are used, or when the physical removal of the 32 bacteria is attempted by oral brushing and rinsing [1-3].” The citation does not concern research that supports this statement. Remove the mentioned agents with the potential of antibiotic or cite original studies that show that the use of lactoferrin lactoperoxidase or lysozyme is ineffective in the plaque model. Line 37 applies to the conversion of a period to a comma between quoted works 7 and 8. Line 39 remove the position 10 of the literature; does not apply to the sentence about which the authors write. Position 11 of the literature is wrongly quoted in the list of references at the end. To the selected fragment, one item of the literature is enough for people, not rats, and is an original and not an illustrative work. Last sentence of introduction „This 68 study suggests that short size DNAs associated with GTF and MVs are important for the formation 69 of biofilms by pathogenic factors and may be a new target” should be rewritten because this is not the first such report. I'm sending you to work „Targeting microbial biofilms: current and prospective therapeutic strategies” Hyun Koo, Raymond N. Allan, Robert P. Howlin, Paul Stoodley and Luanne Hall-Stoodley, who are writing about it and giving a number of original studies on this topic. In the results to present photos with better resolution, maybe the solution was to put them in supplementary files. In this form, some results can be considered as speculation. Besides the Ms is interestingly presented.

Author Response

Responses to comments

Introduction: Rewrite the sentence: „when various antibacterial agents, 31 such as lactoferrin, lactoperoxidase, and lysozyme, are used, or when the physical removal of the 32 bacteria is attempted by oral brushing and rinsing [1-3].” The citation does not concern research that supports this statement. Remove the mentioned agents with the potential of antibiotic or cite original studies that show that the use of lactoferrin lactoperoxidase or lysozyme is ineffective in the plaque model.

Answer) I agreed reviewer’s comments about description of this sentence. “when various antibacterial agents, such as lactoferrin, lactoperoxidase, and lysozyme, are used, or” has been deleted.

Line 37 applies to the conversion of a period to a comma between quoted works 7 and 8.

Answer) I have changed “.” to “,”.

Line 39 remove the position 10 of the literature; does not apply to the sentence about which the authors write.

Answer) I have omitted reference no. 10.

Position 11 of the literature is wrongly quoted in the list of references at the end. To the selected fragment, one item of the literature is enough for people, not rats, and is an original and not an illustrative work.

Answer) Thank you for your suggestion about references. I agreed reviewer’s comments and omitted reference no. 11 and no. 13 in a submitted manuscript.

Last sentence of introduction „This 68 study suggests that short size DNAs associated with GTF and MVs are important for the formation 69 of biofilms by pathogenic factors and may be a new target” should be rewritten because this is not the first such report. I'm sending you to work „Targeting microbial biofilms: current and prospective therapeutic strategies” Hyun Koo, Raymond N. Allan, Robert P. Howlin, Paul Stoodley and Luanne Hall-Stoodley, who are writing about it and giving a number of original studies on this topic. In the results to present photos with better resolution, maybe the solution was to put them in supplementary files. In this form, some results can be considered as speculation. Besides the Ms is interestingly presented.

Answer) Thank you for your suggestion about my conclusion. This comment is very important for our manuscript. I have referred Dr. Koo’s paper and added some sentences in P. 24 line 438-444. P. 2 line 42, Introduction, P. 4 line 93, P.24 line 443 and P. 25 line 457, “new target” has been changed to “one of additional targets”.

English spell and grammars were checked by a English speaker and revised carefully.
